# Human Biomonitoring Data in Health Risk Assessments Published in Peer-Reviewed Journals between 2016 and 2021: Confronting Reality after a Preliminary Review

**DOI:** 10.3390/ijerph19063362

**Published:** 2022-03-13

**Authors:** Tine Bizjak, Marco Capodiferro, Deepika Deepika, Öykü Dinçkol, Vazha Dzhedzheia, Lorena Lopez-Suarez, Ioannis Petridis, Agneta A. Runkel, Dayna R. Schultz, Branko Kontić

**Affiliations:** 1Department of Environmental Sciences, Jožef Stefan Institute, Jamova Cesta 39, 1000 Ljubljana, Slovenia; agneta.runkel@ijs.si (A.A.R.); branko.kontic@ijs.si (B.K.); 2Jožef Stefan International Postgraduate School, Jamova Cesta 39, 1000 Ljubljana, Slovenia; 3Department of Environmental Chemistry, Institute of Environmental Assessment and Water Research (IDAEA), Spanish Council for Scientific Research (CSIC), Jordi Girona, 18, 08034 Barcelona, Spain; marco.capodiferro@idaea.csic.es; 4Departament d’ Enginyeria Quimica, Universitat Rovira i Virgili, Av. Països Catalans 26, 43007 Tarragona, Spain; deepika@urv.cat; 5Department of Physiology and Pharmacology “Vittorio Erspamer”, Sapienza Università di Roma, Piazzale Aldo Moro 5, 00185 Rome, Italy; oyku.dinckol@iss.it; 6Centre for Behavioural Sciences and Mental Health, Istituto Superiore di Sanità, Viale Regina Elena 299, 00161 Rome, Italy; 7Environmental Engineering Laboratory, Department of Chemical Engineering, Aristotle University of Thessaloniki, 54124 Thessaloniki, Greece; vazhajejeya@gmail.com (V.D.); ioannis.petridis89@gmail.com (I.P.); daynaraeschultz@gmail.com (D.R.S.); 8Inserm UMR S-1124, Université de Paris, T3S, F-75006 Paris, France; lorena.lopez_suarez@etu.parisdescartes.fr; 9HERACLES Research Center on the Exposome and Health, Center for Interdisciplinary Research and Innovation, Balkan Center, Bldg. B, 10th Km Thessaloniki-Thermi Road, 57001 Thermi, Greece

**Keywords:** review, human biomonitoring, health risk assessment, exposure assessment

## Abstract

Human biomonitoring (HBM) is a rapidly developing field that is emphasized as an important approach for the assessment of health risks. However, its value for health risk assessment (HRA) remains to be clarified. We performed a review of publications concerned with applications of HBM in the assessment of health risks. The selection of publications for this review was limited by the search engines used (only PubMed and Scopus) and a timeframe of the last five years. The review focused on the clarity of 10 HRA elements, which influence the quality of HRA. We show that the usage of HBM data in HRA is limited and unclear. Primarily, the key HRA elements are not consistently applied or followed when using HBM in such assessments, and secondly, there are inconsistencies regarding the understanding of fundamental risk analysis principles and good practices in risk analysis. Our recommendations are as follows: (i) potential usage of HBM data in HRA should not be non-critically overestimated but rather limited and aligned to a specific value for exposure assessment or for the interpretation of health damage; (ii) improvements to HRA approaches, using HBM information or not, are needed and should strictly follow theoretical foundations of risk analysis.

## 1. Introduction

Human biomonitoring (HBM) refers to measuring the presence and levels of substances in different human tissues (hair, blood, urine, etc.). Measured biomarkers are either markers of exposure or of an effect and provide aggregated information about different exposures through different pathways [1]. Despite confirming that an exposure occurred, the exposure biomarkers are actually direct measurements of a dose and not exposure. The differences between the two terms need to be acknowledged for appropriate HBM data interpretation within the environmental health paradigm [2] and especially in terms of exposure assessment for risk-assessment purposes (more on this issue is in the discussion section). Health risk assessment (HRA) is a method that uses “factual base to define the health effects of exposure of individuals or population to hazardous materials and situations” [3]. General principles and fundamental elements of HRA were established by the risk assessment “Red book” [3] and continue to form the basis of developing HRA, despite being the subject of extensive discussion in various notable publications since then [4,5,6,7]. HRA should not be viewed as an end in itself, but as a method for evaluating the relative merits of various risk-management options [6] and has been used to inform various decision makers in protecting human health and the environment from a range of threats [5]. HBM and HRA present potential for addressing environmental health and public health concerns. HBM unequivocally confirms whether individuals or populations have been exposed and can, when used with available epidemiologic [8], toxicological [9], and pharmacokinetic (modeling) data [10], help in the estimation of the amount of substance absorbed into the body, which could indicate potential health risks [11]. HBM can improve estimates of exposure and dose [12] and has been continuously emphasized to potentially improve HRA for both workers and the general population [13,14,15,16].

A 2006 publication by the National Research Council (NRC) identified only a few HRA cases based on biomarker-response relationships established in epidemiologic studies and noted that, despite the potential presented in HBM information, it only rarely reduced uncertainty in the practice of HRA [11]. More recent publications, checked randomly [17,18,19], do not report a change in this NRC observation. The analysis of the number of documents by year shows that the number of publications in the HBM area has been rising substantially since around 2006 (Figure 1). A similar trend can be observed in the number of documents per year published about both HBM and HRA.

The interest in systematically checking the recent situation regarding the practice and usefulness of HBM data in HRA led us to design a review study of selected peer-reviewed publications published in the last five years (between January 2016 and April 2021). This review aimed to address two main questions:Are fundamental elements of HRA [5] considered in the publications on the practical integration of HBM data and HRA?In which HRA elements is the use of HBM data clearly demonstrated and reported?

This study also aimed to re-assess the validity of the observation by the NRC from 2006 that “the ability to generate new biomonitoring data often exceeds the ability to evaluate whether and how a chemical measured in an individual or population may cause a health risk or to evaluate its sources and pathways for exposure” [11] (p. 2).

## 2. Materials and Methods

### 2.1. Publication Search

The identification of peer-reviewed publications for the subject review matched the following criteria: publications had to involve both “human biomonitoring” and “risk assessment” in their title, keywords, or abstract and had to be published in the last five years. The PubMed (https://pubmed.ncbi.nlm.nih.gov/) and Scopus (https://www.scopus.com/) search engines were used. The publication search was performed on 30 April 2021. The following search queries were applied:PubMed: (((“risk assessment” [Title/Abstract] OR “HRA” [All Fields]) AND (“HBM” [Title/Abstract] OR “human biomonitoring” [Title/Abstract])) AND (y_5[Filter])) NOT (review [Title/Abstract]) Filters: in the last 5 yearsScopus: (TITLE-ABS-KEY (“risk assessment” OR hra) AND TITLE-ABS-KEY (“Human biomonitoring” OR hbm) AND NOT TITLE-ABS-KEY (review)) AND (LIMIT-TO (PUBYEAR, 2021) OR LIMIT-TO (PUBYEAR, 2020) OR LIMIT-TO (PUBYEAR, 2019) OR LIMIT-TO (PUBYEAR, 2018) OR LIMIT-TO (PUBYEAR, 2017) OR LIMIT-TO (PUBYEAR, 2016)) AND (LIMIT-TO (DOCTYPE, “ar”)) AND (LIMIT-TO (EXACTKEYWORD, “Risk Assessment”)).

The search queries returned 83 records on PubMed and 140 on Scopus (Figure 2). Records of both databases were collated in Mendeley reference manager (https://www.mendeley.com, accessed on 11 March 2022). After the removal of the duplicates (*n* = 56), the remaining 167 records underwent eligibility screening. Eligibility assessments were performed by reviewing their titles, keywords, and abstracts based on the pre-defined eligibility criteria: articles had to focus on specific populations and the estimation/assessment/calculation/characterization of health risks in the selected population; however, review publications or method development publications were excluded. In total, 36 publications were selected, successfully retrieved, and included in the appraisal.

### 2.2. Appraisal Tool

A review of the presence and clarity of different fundamental HRA elements in publications about HBM and the assessments of health risks was performed with the help of a straightforward and transparent appraisal tool (see Appendix A). It was designed for this particular review purpose and covers the evaluation of 10 selected HRA elements, namely the assessment context of HRA, dose/exposure-response relationship, exposure setting, exposure sources, exposure duration, exposed population, magnitude of risk, uncertainty of HRA results, options for mitigating/avoiding exposure, and transparency and clarity of the assessment process. These HRA elements are consistent with the core principles of HRA and risk analysis [3,6,12] and are among the proposed key elements for judging the quality of HRA [22]. The evaluation was performed by using a straightforward questionnaire and was, overall, both limited and preliminary. We intend to repeat the evaluation in the following years by involving a larger number of experienced experts and specialists in both fields of HBM and HRA.

The appraisal tool consisted of 10 appraisal questions about selected HRA elements (Table 1). Each of the selected HRA elements has various important aspects discussed in detail elsewhere [3,5,23,24]. To improve objectivity, the appraisal questions narrowed the focus regarding each HRA element and facilitated a clear “Yes” answer if the publication demonstrated that the HRA element had been clearly applied/reported (or “No” if it did not) and an “X” mark if it was clearly demonstrated that HBM data were used in a specific HRA element (if not, the column was left blank). Comments provided additional clarifications when appropriate. There were multiple discussions regarding the clarity of the appraisal tool among all persons involved in the review before and during the evaluation to improve the consistency of the review findings. The evaluation was performed by 10 of 14 NEUROSOME early stage researchers (ESRs) from different backgrounds and areas of interest between May and August 2021. NEUROSOME is a Horizon 2020 funded integrated training network that investigates causal associations among genetic predispositions, exposures to multiple environmental chemicals and neurodevelopmental disorders according to the exposome paradigm [25]. Within NEUROSOME [26] the ESRs conducted research in the leading research institutions in France, Greece, Italy, Slovenia, and Spain, and participated in different training activities and courses, which among other matters covered various aspects of HBM measurements and HBM data interpretation as well as selected topics related to different elements of HRA (e.g., hazard evaluation, dose–response evaluation, exposure assessment). Therefore, they were considered competent enough for reviewing selected publications. Every ESR appraised at least two publications that were related to their area of research and expertise as much as possible. Despite multiple discussions during the appraisal, there was no appraisal of a single publication by two or more ESRs in order to avoid the demanding step of harmonizing potential differences in their findings. Such an organization of the evaluation also contributed to its preliminary nature, which intentionally had a limited scope; the aim was to illustrate only the most general understanding of the covered topics among professionals with different backgrounds. A more comprehensive review with a wider scope and including more relevant professionals will follow this preliminary review.

## 3. Results

The results and discussion are presented as a summary of the general findings. Sub-sections discuss conclusions regarding the dose/exposure-response and exposure assessment, the overall HRA process, its results and risk management, the value of HBM data for HRA and risk management, and the strengths and weaknesses of the study.

The limited review found that although the appraised papers reported some type of assessment of risks, with some claiming to perform an HRA, none of them evaluated all of the HRA elements included in the appraisal or provided an argument for why these elements were not addressed. Furthermore, the review of the publications did not provide any clear conclusions regarding the actual usefulness of HBM information within the risk analysis context—for the HRA, risk communication, and especially for risk management purposes. None of the HRA elements were included and assessed as clear for any of the appraised publications (Table 2). Most of the appraised publications did not clearly demonstrate the use of HBM for any of the HRA elements. The majority of the appraised publications were, despite stating otherwise in their abstracts, titles, or keywords, not actual examples of (comprehensive) HRAs, but were rather HBM-based exposure assessment studies, which, while undeniably confirming that the exposure to a detected substance or its metabolite occurred, lacked clear information regarding the other important exposure assessment estimates stressed by Sexton et al. [12]: for instance, activities causing the exposure, exposure sources, pathways, population exposed, etc. The limited assessment of risks in the studies was mostly performed through various types of threshold value approaches, such as comparisons with guidance values, acceptable daily intake values, reference doses, etc. The observation of the NRC that the ability to generate new HBM data exceeds the ability to assess whether and how a substance measured in an individual or population can cause health risks, or to evaluate exposure sources and pathways seems to be as strong and relevant as it was 15 years ago [11]. The Appendix A includes a collection of comments accompanying the responses collected in Table 2.

## 4. Discussion

### 4.1. Dose/Exposure-Resopnse and Exposure Assessment

The dose/exposure-response relationship was evaluated as clear in 33% of the appraised publications (Table 2). Similarly, 31% of the appraised publications clearly demonstrated the use of HBM for the dose/exposure-response element of HRA. This finding is not surprising, since the HBM studies mostly assessed risks using one of the threshold value-based approaches, which only compare the estimated HBM-derived exposure estimates with various guidance values (regulatory limits, tolerable daily intakes, acceptable daily intakes, etc.). Authoritative bodies place too much focus on human health reference values [63] and continue to promote threshold-value based types of HRA results [23,64], which are—as observed in the reviewed studies—often reported as the only measures of risk without clear reporting and a discussion of the strength of knowledge and assumptions behind the specific guidance value and without the applicability of the selected HRA approach in each specific case. Despite being based on actual exposure/dose–response information, the threshold-value based approaches lead to under-acknowledgment of the “dose makes the poison” principle [65] across the entire range of possible exposures/doses and are limited in their ability to account for potentially important individual susceptibilities in the population of interest.

An exposure assessment involves the evaluation of the exposure of an organism or group of organisms [66] along with the characteristics of those exposed. It should ideally describe the exposure sources, pathways, routes, and uncertainties in the assessment [67]. An exposure assessment is the most critical step in the process of HRA since, without exposure, there is no risk or related adverse health effects. In the HRA process, the exposure assessment is usually the key area of uncertainty [24]. Our review included only four of the many important features of an in-depth, comprehensive exposure assessment. Additional exposure assessment elements that have not been included are, for example, exposure route, exposure point, exposure concentration, relevant environmental characteristics, etc. The exposure setting was evaluated as clear in 36% of the appraised publications, exposure sources were evaluated as clear in 42% of the appraised publications, and exposed populations were evaluated as clear in 31% of the appraised publications (Table 2). Exposure duration (8%) was one of the HRA elements that were evaluated as the least clear or not included in the appraised publications.

The use of HBM for evaluating exposure setting and exposure duration was clearly demonstrated in only four publications, the use of HBM for assessing exposure sources was clear in five publications, and 11 publications demonstrated a use of HBM for the assessment of the exposed population (Table 2). HBM demonstrates that exposure and uptake have occurred, but only provides direct information about internal presence and concentration (and rarely about dose) that is integrated across all types of exposure routes; it usually does not provide information about the relative importance of inhalation, ingestion, and dermal absorption. Serious limitations when reconstructing exposure based on HBM data include a lack of physiologically based pharmacokinetic models, an underlying lack of good understanding of pharmacokinetics, a lack of data for exposure situations, unvalidated default assumptions, etc. The papers reviewed did not provide clear answers to the majority of questions that should be considered when designing, conducting, or interpreting exposure studies in the context of biomonitoring, such as “have the primary sources of exposure been identified?”, “are the pathways/routes of exposure understood?”, “can human exposure be related to animal toxicology studies?”, “is there some understanding of the exposure-dose relationship?”, and “what is understood about temporality and duration of exposure?” [14] (p. 1758). 

The risk analysis area is riddled with foundational issues that include an inconsistent understanding and acknowledgment of its main concepts and principles [68,69]. We can confirm that the confusion about terminology that seems to persist as one of the major problems of HRA [70,71] is also found in the area of HBM, as indicated by the studies included in our review. One such instance of confusion is related to the use of “internal exposure”. HBM information is often reported as a measure of internal exposure [13,55,72,73]. While internal exposure is distinguished from external exposure in the case of radiation exposure [74], the difference between the two is in whether the source that emits radiation lies inside or outside the body, which is not applicable for nonradioactive substances. Without a clear meaning for the term, the use of “internal exposure” creates confusion, especially if established definitions of “exposure” and “dose” are considered (see Table 3). We argue that exposure biomarkers are, in general, direct or indirect measurements of a dose and that there is no need for the introduction and use of the term “internal exposure.” The use of “internal exposure” does not contribute to clarity regarding the value of HBM for exposure assessment and HRA, and it is confusing when placing the HBM information within the environmental public health paradigm, which covers multiple areas, starting from the release of a substance (i.e., sources) to the adverse health outcomes in individuals or populations [2,75]. A recognized need for a better assessment of the link between external exposure sources and internal exposure [13] additionally illustrates the unnecessary use of “internal exposure” instead of “dose.”

### 4.2. Process and Results of Health Risk Assessment, and Risk Management

HRA needs to strive for transparency and clarity, in the same way as any form of scientific research [77]. Although none of the reviewed publications reported a comprehensive HRA, the assessment process was evaluated as transparent and clear in two-thirds of them, while only five publications clearly demonstrated the importance of HBM in the overall transparency and clarity of the publication (Table 2).

All persons involved in the HRA process and the users of HRA results, such as policy makers, public, etc., can come from various backgrounds, and can have different needs and expectations regarding the HRA. To ensure the utility of HRA results for specific risk-informing purposes, a consensus regarding the terminology, concepts and methods used in specific HRA needs to be reached. The HRA context must be clarified among all relevant stakeholders involved in the HRA process in its early stages. Clarification of the HRA context should provide clear answers to the questions “What is to be assessed?” and “Why is it to be assessed?”; such answers should be in accordance with future risk management decisions. During the assessment context step, all involved parties need to contribute to the clarity of the decision and assessment problem, scope of the assessment, methods to be applied, and available resources, including time constraints, etc. [5,78]. However, this preliminary review revealed that the assessment context was perceived as clear in only 44% of the publications (Table 2). The value of HBM for the assessment context was demonstrated clearly in 42% of the appraised publications (Table 2). Since none of the papers reported a comprehensive example of HRA, our review could not clearly distinguish between the context of the respective study and the context of the actual HRA, which may not be the same.

In general, the HRA process aims to assess the magnitude of risk (i.e., the severity of consequences), its probability, and the strength of knowledge supporting the assessment findings, which includes the uncertainty assessment [7]. Considering all of the above, the assessment of risk performed only with a comparison with guidance values is limited. This may explain why the magnitude of risk was understood among the HRA elements that were the least clear or not included in the appraised publications (Table 2). Only three publications clearly demonstrated the use of HBM for the assessment of the magnitude of risk. The uncertainty of the HRA results was clear in 56% of reviewed publications (Table 2). However, several publications (see Appendix A) only provided a general uncertainty assessment or limitations assessment of the entire study. The use of HBM for assessing the uncertainty of HRA results was clear in only three publications.

From the risk-informed decision-making point of view, alternative decisions and/or options for mitigating or avoiding exposure are among the most important HRA elements. In specific cases, e.g., in the case of flame retardants [79], risk management must weigh the costs and benefits of various options. Our review showed that the options for mitigating exposure were among the HRA elements that were the least clear or not included in the appraised publications (Table 2). Only two publications demonstrated the use of HBM for assessing or identifying options for mitigating exposure.

### 4.3. Value of Human Biomonitoring Data for Health Risk Assessment and Risk Management

The observations of our review are in line with the conclusions of a review of the state-of-the-art use of HBM in HRA in Europe. It suggests that significant work is still needed to improve the implementation of HBM in regulatory HRA [13]. Figure 3 illustrates potential uses of HBM in the health risk assessment and risk-analysis contexts. The exposure assessment is a crucial element of HRA, especially in terms of identifying potential risk-management options. HBM can provide robust proof that an exposure to a certain substance or stressors has occurred (exposure biomarkers), can inform the assessor about specific health effects (effect biomarkers), or can be suitable for the development of a mechanistic understanding of environmental health processes (Figure 3 points 4 and 5). From a risk management perspective, it is essential to link biomarkers to exposure-related events, whereby public or private actions and changes in lifestyle can reduce the probability of adverse health outcomes. The value of biomarkers for exposure assessment “depends on whether they can be used to reconstruct internal dose and related exposures, and on whether they aid in identifying and quantifying the relative contributions of various sources and pathways to exposure/dose” [12] (p. 25). HBM often does not reveal exposure sources and routes [80] and even when the distribution of biomarkers of exposure or effect is well characterized in a defined population, and when there is a solid understanding of exposure routes and contributing sources, it remains challenging to predict the influence of changes in emissions from a small number of identified sources on the distribution of biomarkers [6]. Technological advances (e.g., high throughput mass spectrometry) have facilitated measurements of a large number of environmental agents. However, the challenge of including biomarkers of exposure and response in the development and validation of specific and sensitive measures of pathway perturbations and environmental exposures still exists [4]. HBM can also be useful for the identification of exposed (susceptible) populations, and can, together with “matching” environmental monitoring, be used for monitoring purposes (e.g., following an implementation of specific decisions, changes in specific activities/interventions, etc.; Figure 3 point 7). The clarification of the fitness and usefulness of HBM for specific HRA purposes requires a clear understanding of HBM information within the environmental health paradigm context. Without it, the knowledge acquired via the HRA process is not complete and cannot provide the best possible information for risk-informed decision making. Direct exposure measurements and measurements of dose (i.e., biomarkers) are not interchangeable but are complementary rather than competing methods for conducting realistic exposure assessments. It is critical to couple the HBM data with the collection of relevant environmental exposure, source, and health data to allow for the best possible interpretation of the implications of exposures to facilitate prevention and intervention [81]. In addition, the exposure information obtained must be accessible and its meaning and limitations made clear to community members if it is to inform decisions involving exposure prevention or intervention (Figure 3 points 2, 3 and 6) [81].

The usefulness of HBM information for specific HRA cannot be expected a priori if the assessment context [78] (a clear definition of what is to be assessed, assessment endpoints, assessment purpose—e.g., decisions about changing a causal relationship, or prevention of exposure, etc.) is not clarified among all relevant stakeholders, and if it is not considered during the planning of HBM and HRA (Figure 3 points 2 and 3). Expectations of obtaining useful HBM information for effective decision making in terms of policy development for the areas and populations of concern are reasonable only if a clear HRA purpose drives the HBM programs. If this is not the case, HBM information may only have limited value in terms of risk management. Furthermore, HBM information may implicate potential relationships between causes and effects, which warrants further investigation and assessment or may be used to document trends and status (e.g., population reference values). Documentation of trends and status is the simplest and least informative way of using HBM information, even if the levels measured can be compared to certain standards for evaluating the level of concern (i.e., hazard index, hazard quotient; Figure 3 point 1) [82]. Such comparisons must not be construed as more comprehensive HRAs with better risk-informed decision-making potential. In the absence of other information, assumptions are inevitable for the statistical associations between the measured concentrations and potential exposure sources identified by questionnaire responses, or for the estimation of exposure routes that can directly determine biologically effective dose, as illustrated by the example of chloroform exposure from showering [83]. The rationale behind the assumptions, which are necessary when the knowledge required for specific HRA is not complete, must be reported clearly. If HBM information is not reasonably fit for the specific HRA purpose, its utility for HRA cannot be improved by questionable assumptions that allocate it an additional desired value. If this is done, caveats stronger than formal uncertainty discussion regarding usage of these data and related HRA results in policy contexts by decision makers must be made. Advances in the HRA and risk analysis areas should acknowledge the need for greater stakeholder participation in both HRA and risk management [84], which makes the “decision process more democratic, improve the relevance and technical quality of the assessment and increases the legitimacy and public acceptance of the resulting decisions” [85] (p. 689). These issues can be addressed by procedural improvements, as emphasized and addressed by the various existing assessment frameworks [5,78]. A consideration of such a framework when performing HRAs can identify and clarify the need for HBM and its value for the assessment.

### 4.4. Strengths and Weaknesses

This review included 36 scientific publications that do not represent the entire body of research and related practice in the areas of HRA and HBM. Since the review only evaluated the presence, transparency, and clarity of the selected HRA elements in the selected publications, it cannot represent the actual understanding of the topics covered among the authors of the publications. There are many other important HRA or risk analysis elements that were not included in the appraisal tool, which does not mean they are without importance in specific studies/assessments; for example, stakeholders’ participation, judgment of the strength of knowledge, peer review, etc., [22,86]. While the appraisal tool and the review did not focus on such elements, some of them may be clearly included in the appraised publications but were not considered in the evaluation.

HRAs are inherently subjective, as “the definition of risk controls the rational solution to the problem at hand” [85] (p. 699). By acknowledging the inherent subjectivity of all involved in the appraisal and the multidisciplinarity of the HRA and risk analysis areas, the appraisal aimed to mimic/represent a “real-world” situation of risk analysis cases involving multiple stakeholders (decision-makers, researchers, the public, etc.). Despite careful preparation of the appraisal tool, we cannot overlook the preliminary and limited character of the appraisal findings; this is also due to the limited experience of the ESRs who performed the review. It is important to note, however, that the limited review was not performed in a way that would force a specific understanding of the HRA elements upon all involved, but instead acknowledged the differences in their understanding. It was based on the assumed capability that those involved could develop a comparable set of criteria to obtain comparable answers to the appraisal questions, despite their different backgrounds, education, interests, and, last but not least, professional beliefs and values [84]. This assumption was confirmed several times during the development of the appraisal tool and during the review through multiple discussions. In this view, simple sums and percentages of specific answers as presented in Table 2 curb the different opinions and findings among the ESRs, if they were all reviewing all articles. Such an approach to the appraisal also avoided the inevitable step of consultation and harmonization among the ESRs about each of the reviewed articles (by applying, e.g., the Delphi method), which would otherwise be necessary. The discussions and the comments, which provided additional argumentation about appraisal findings, indicated that the inherent and inevitable differences in the understanding of HRA elements and especially in the understanding of papers reviewed potentially lead to only minor differences in the appraisal findings, which did not affect the general findings of the study. Nevertheless, our findings can inform future developments of the interconnected areas of HRA and HBM.

## 5. Conclusions

The application of HRA theory (e.g., its terminology and concepts) and practice in the human biomonitoring area is not consistent. While HBM has advantages, primarily as an undeniable proof of exposure, it has limited value in providing other types of crucial exposure-assessment information when assessing risks (i.e., exposure sources, exposure pathways, why are individuals/population exposed, etc.) and for targeted risk management interventions. Many of the HBM studies did not thoroughly specify the underlying uses and usefulness of HBM data for HRA purposes before sample collection. This leads to increasing amounts of HBM information that remain archived but unexploited in terms of their expected, even promised, yet unrealized usefulness for HRA and related risk-informed decision making.

The following points need to be considered to improve the risk-informing potential of HRAs that use HBM data:Stakeholder involvement in the early stages of HRA is crucial for the clarification of an assessment context. Clear assessment context assures that HRA can address the needs or concerns of decision makers or other stakeholders. HBM, if performed for the purpose of HRA, must acknowledge the assessment context in its planning stages.The lack of stakeholder involvement (e.g., when using existing databases) needs to be reported along with the discussion about the usefulness of obtained HRA results for specific purposes.The use of the term “risk assessment” creates confusion/false expectations among decision makers or other stakeholders if only parts of the HRA process are practiced.Underlying assumptions of HRA (e.g., related to HBM based exposure assessment, lacking pharmacokinetic knowledge, etc.) must be reported and thoroughly discussed, since they can be an important source of uncertainty or study limitations.

## Figures and Tables

**Figure 1 ijerph-19-03362-f001:**
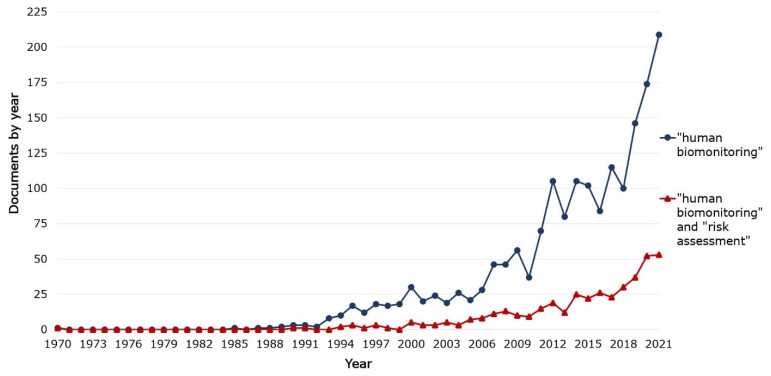
Documents by year for two types of searches using keywords on Scopus: “human biomonitoring” and “human biomonitoring” and “risk assessment” [20,21].

**Figure 2 ijerph-19-03362-f002:**
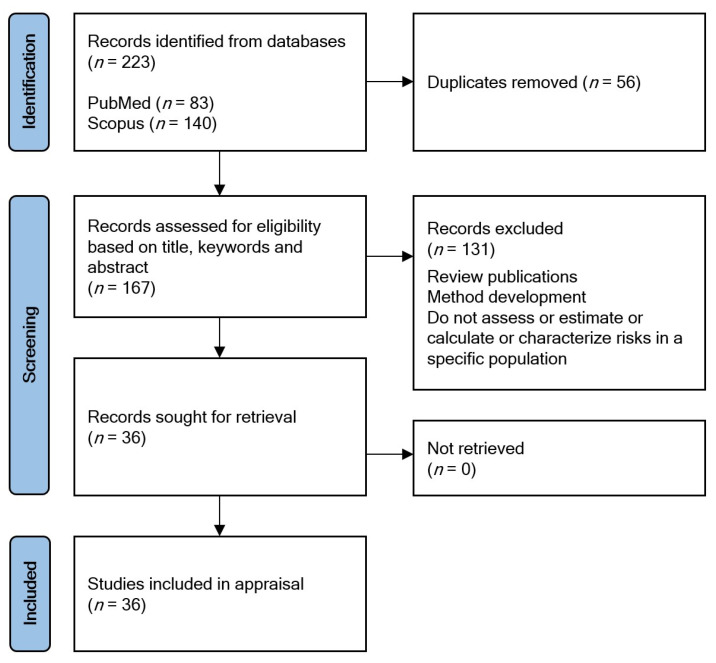
PRISMA flow diagram.

**Figure 3 ijerph-19-03362-f003:**
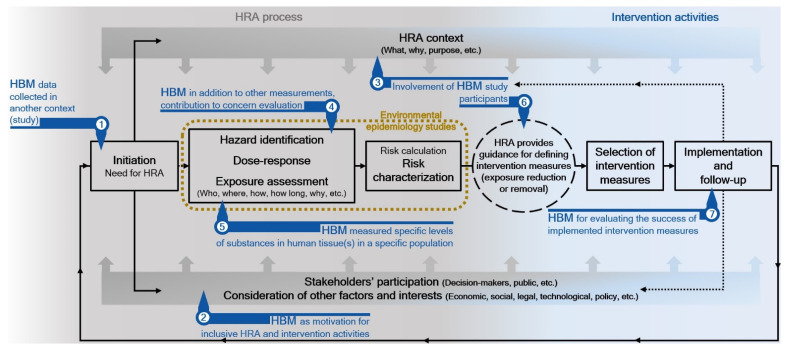
Potential uses of human biomonitoring in the health risk assessment context.

**Table 1 ijerph-19-03362-t001:** Selected health risk assessment (HRA) elements for the appraisal and related appraisal questions.

Appraised HRA Element *	Appraisal Question
Assessment context of HRA	Does the assessment clearly identify what is assessed and why at the start? Has assessment context been followed/applied in the HRA process?
Dose/exposure—response relationship	Is the applicability of the selected dose/exposure-response relationship for the assessment thoroughly discussed?
Exposure setting	Are the characteristics of the place of exposure clearly described?
Exposure sources	Are the major sources of hazardous material and/or activities causing the release(s) of hazardous material(s) into the environment identified?
Exposure duration	Is the duration and frequency of the exposure identified?
Exposed population	Is it clear who is really exposed (population/individuals, their number), and why are they exposed (e.g., their activities leading to exposure)?
Magnitude of risk	Are the types of the expected adverse outcomes, their severity and the probability of their occurrence identified clearly?
Uncertainty of HRA results	Are the major sources of uncertainty evaluated?
Options for mitigating/avoiding exposure	Are there any specific actions for avoiding or mitigating the exposure to the selected hazardous materials identified and/or proposed?
Transparency and clarity of the assessment process	Is it transparent and clear how was the assessment performed and its conclusions obtained?

* A more complete list and description of HRA process and all of its elements can be found elsewhere [3,5,6].

**Table 2 ijerph-19-03362-t002:** Clarity of HRA elements (Yes or No) and the use of human biomonitoring in specific HRA elements (marked with X).

Publication Title	Assessment context of HRA *	Dose/exposure—Response	Exposure Setting	Exposure Sources	Exposure Duration	Exposed Population	Magnitude of Risk	Uncertainty of HRA Results	Options for Mitigating Exposure	Transparency and Clarity
1. Biomonitoring and health risks assessment of trace elements in various age- and gender-groups exposed to road dust in habitable urban-industrial areas of Hefei, China [27]	No	No	Yes	No	No	Yes	No	No	Yes	Yes
	X									
2. Health Risk Assessment of Trace Metals Through Breast Milk Consumption in Saudi Arabia [28]	Yes	No	No	Yes	No	Yes	Yes	Yes	No	Yes
	X						X			
3. Exposure levels, determinants and risk assessment of organophosphate flame retardants and plasticizers in adolescents (14–15 years) from the Flemish Environment and Health Study [29]	No	No	Yes	Yes	No	No	No	Yes	No	Yes
	X									
4. Organophosphate pesticide exposure in children in Israel: Dietary associations and implications for risk assessment [30]	No	No	No	No	No	No	No	Yes	Yes	Yes
						X			X	
5. Exposure of Portuguese children to the novel non-phthalate plasticizer di-(iso-nonyl)-cyclohexane-1,2-dicarboxylate (DINCH) [31]	No	No	No	No	No	No	No	No	No	No

6. Exposure and Risk Assessment of Hg, Cd, As, Tl, Se, and Mo in Women of Reproductive Age Using Urinary Biomonitoring [32]	No	No	No	No	No	No	No	No	No	No

7. Exposure and risk assessment of the Czech population to chlorinated pesticides and polychlorinated biphenyls using archived serum samples from the period 1970 to 1990 [33]	Yes	Yes	Yes	Yes	No	No	No	Yes	No	Yes
	X	X	X	X	X	X				X
8. Risk assessment of deoxynivalenol in high-risk area of China by human biomonitoring using an improved high throughput UPLC-MS/MS method [34]	No	No	No	No	No	No	No	No	No	Yes
						X				
9. Risk assessment of exposure to phthalates in breastfeeding women using human biomonitoring [35]	Yes	Yes	No	Yes	No	Yes	No	Yes	No	Yes
	X	X		X		X				
10. Evaluation of human biomonitoring data in a health risk based context: An updated analysis of population level data from the Canadian Health Measures Survey [36]	No	No	No	No	No	No	No	Yes	No	Yes
		X				X		X		
11. Biomonitoring of non-persistent pesticides in urine from lactating mothers: Exposure and risk assessment [37]	No	No	No	No	No	No	No	No	No	No

12. Children’s exposure to polycyclic aromatic hydrocarbons in the Valencian Region (Spain): Urinary levels, predictors of exposure and risk assessment [38]	Yes	No	Yes	Yes	Yes	Yes	Yes	No	No	Yes
			X	X	X	X	X			
13. Evaluation of exposure to phthalate esters and DINCH in urine and nails from a Norwegian study population [39]	Yes	Yes	No	Yes	No	Yes	No	No	No	Yes
	X	X				X				
14. Wastewater-based epidemiology for tracking human exposure to mycotoxins [40]	No	No	No	Yes	No	No	No	Yes	No	Yes
						X				
15. Biomonitoring of polychlorinated dibenzo-p-dioxins (PCDDs), polychlorinated dibenzofurans (PCDFs) and dioxin-like polychlorinated biphenyls (dl-PCBs) in human milk: Exposure and risk assessment for lactating mothers and breastfed children from Spain [41]	No	Yes	No	No	No	No	No	Yes	No	No
		X								
16. Predicted Mercury Soil Concentrations from a Kriging Approach for Improved Human Health Risk Assessment [42]	Yes	No	Yes	Yes	No	Yes	No	No	No	Yes
	X			X						
17. Lead and mercury levels in repeatedly collected urine samples of young children: A longitudinal biomonitoring study [43]	No	No	No	No	No	No	No	Yes	No	No
								X		
18. Exposure to the plasticizer di(2-ethylhexyl) terephthalate (DEHTP) in Portuguese children–Urinary metabolite levels and estimated daily intakes [44]	Yes	Yes	Yes	No	No	Yes	No	No	No	Yes
	X	X								X
19. Exposure and health risk assessment of secondary contaminants closely related to brominated flame retardants (BFRs): Polybrominated dibenzo-p-dioxins and dibenzofurans (PBDD/Fs) in human milk in Shanghai [45]	No	No	No	No	No	No	No	No	No	No

20. Integration of biomonitoring data and reverse dosimetry modeling to assess population risks of arsenic-induced chronic kidney disease and urinary cancer [46]	Yes	Yes	Yes	Yes	No	Yes	Yes	Yes	Yes	Yes
	X	X								
21. Exposure assessment of Portuguese population to multiple mycotoxins: The human biomonitoring approach [47]	No	No	No	No	No	No	No	Yes	No	Yes
								X		X
22. Glyphosate in Portuguese Adults- A Pilot Study [48]	No	No	No	No	No	No	No	Yes	No	No

23. Exposure of nursing mothers to polycyclic aromatic hydrocarbons: Levels of un-metabolized and metabolized compounds in breast milk, major sources of exposure and infants’ health risks [49]	Yes	No	Yes	No	Yes	Yes	No	No	No	Yes
	X									
24. Biomonitoring of mercury in hair of children living in the Valencian Region (Spain). Exposure and risk assessment [50]	No	No	No	No	No	No	No	No	Yes	No
									X	
25. Estimating human exposure to pyrethroids’ mixtures from biomonitoring data using physiologically based pharmacokinetic modeling [51]	Yes	Yes	Yes	No	No	Yes	No	Yes	No	Yes
	X	X								
26. Cadmium exposure in First Nations communities of the Northwest Territories, Canada: smoking is a greater contributor than consumption of cadmium-accumulating organ meats [52]	Yes	Yes	Yes	Yes	No	Yes	No	Yes	No	Yes
		X								
27. Implementation of human biomonitoring in the Dehcho region of the Northwest Territories, Canada (2016–2017) [53]	Yes	Yes	Yes	Yes	No	No	No	Yes	No	Yes
	X		X	X	X	X				X
28. Assessment of human exposure to selected pesticides in Norway by wastewater analysis [54]	No	No	No	Yes	No	No	No	Yes	No	Yes
						X				
29. Biomonitoring of bisphenols A, F, S and parabens in urine of breastfeeding mothers: Exposure and risk assessment [55]	Yes	No	No	Yes	No	Yes	No	No	No	Yes
						X				
30. Integrated exposure and risk characterization of bisphenol-A in Europe [56]	No	No	No	No	No	No	No	No	No	No
			X		X					
31. Risk characterization of bisphenol-A in the Slovenian population starting from human biomonitoring data [57]	Yes	Yes	Yes	No	Yes	No	No	Yes	No	Yes
	X									
32. Human biomonitoring in urine samples from the Environmental Specimen Bank reveals a decreasing trend over time in the exposure to the fragrance chemical lysmeral from 2000 to 2018 [58]	Yes	Yes	No	Yes	No	Yes	No	Yes	No	Yes
		X				X				X
33. Bisphenol A and six other environmental phenols in urine of children and adolescents in Germany-human biomonitoring results of the German Environmental Survey 2014–2017 (GerES V) [59]	Yes	No	Yes	Yes	No	Yes	No	No	No	Yes
	X									
34. Multicenter biomonitoring of polybrominated diphenyl ethers (PBDEs) in colostrum from China: Body burden profile and risk assessment [60]	No	No	No	No	No	No	No	Yes	No	No

35. Biomonitoring and Subsequent Risk Assessment of Combined Exposure to Phthalates in Iranian Children and Adolescents [61]	No	Yes	No	No	No	No	No	No	No	No
		X								
36. Antibiotic body burden of elderly Chinese population and health risk assessment: A human biomonitoring-based study [62]	No	No	No	No	No	No	Yes	Yes	Yes	No
	X						X			
**Number of “Yes” (Yes proportion)**	16 (44%)	12 (33%)	13 (36%)	15 (42%)	3 (8%)	14 (39%)	4 (11%)	20 (56%)	5 (14%)	24 (67%)
**Number of “X” (X proportion)**	15 (42%)	11 (31%)	4 (11%)	5 (14%)	4 (11%)	12 (33%)	3 (8%)	3 (8%)	2 (6%)	5 (14%)

* Assessment context answers the following key questions: what is to be assessed, why is to be assessed, which assessment endpoint is relevant, assessment timeframe; it is more specific than the general context of the publication.

**Table 3 ijerph-19-03362-t003:** Definitions for “exposure” and “dose”.

Term	Definitions
Exposure	“Concentration or amount of a particular agent that reaches a target organism, system, or (sub)population in a specific frequency for a defined duration” [66] (p. 12).
	“Contact between an agent and a target. Contact takes place at an exposure surface over an exposure period” [67] (p. 3).
	1. “Concentration, amount, or intensity of a particular physical or chemical agent or environmental agent that reaches the target population, organism, organ, tissue or cell, usually expressed in numerical terms of substance concentration, duration, and frequency (for chemical agents and micro-organisms) or intensity (for physical agents such as radiation). 2. Process by which a substance becomes available for absorption by the target population, organism, organ, tissue or cell, by any route” [76] (p. 2047).
	“Exposure is defined as contact of a biologic, chemical, or physical agent with the outer part of the human body, such as the skin, mouth, or nostrils” [12] (p. 17).
Dose	“Total amount of an agent administered to, taken up by, or absorbed by an organism, system, or (sub)population” [66] (p. 11).
	“The amount of agent that enters a target after crossing an exposure surface. If the exposure surface is an absorption barrier, the dose is an absorbed dose/uptake dose; otherwise it is an intake dose” [67] (p. 3).
	“Total amount of a substance administered to, taken or absorbed by an organism” [76] (p. 2039).
	“Once the agent enters the body by either intake or uptake, it is described as a ‘dose’” [12] (p. 19). “Potential, or administered dose, is the amount of the agent that is actually ingested, inhaled, or applied to the skin” [12] (p. 19). “Applied dose is the amount of the agent directly in contact with the body’s absorption barriers, such as the skin, respiratory tract, and gastrointestinal tract, and therefore available for absorption” [12] (p. 19.). “The amount of the agent absorbed, and therefore available to undergo metabolism, transport, storage, or elimination, is referred to as the ‘internal’ or ‘absorbed dose’” [12] (p. 19). The portion of the internal (absorbed) dose that reaches a tissue of interest is called the ‘delivered dose’” [12] (p. 19). “The portion of the delivered dose that reaches the site or sites of toxic action is called the ‘biologically effective dose’” [12] (p. 19).

## Data Availability

Not applicable.

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
