# Peer review of "Human Biomonitoring Data in Health Risk Assessments Published in Peer-Reviewed Journals between 2016 and 2021: Confronting Reality after a Preliminary Review"

_ijerph, 2022, doi:10.3390/ijerph19063362_

Round 1

Reviewer 1 Report

This is a novel work and should be a flagship work in the field despite earlier work by the NRC. I am yet to read many publications in this field. I have made some minor changes and restructuring in how the paper can gain more acceptance. Authors should adhere to these changes or provide reasons not to do them. 

Reviewer 2 Report

The paper focuses on the usage of the human biomonitoring data in the health risk assessment context and is an analysis of related publications. It takes into account 36 publications selected from PuBMed and Scopus database. Mendeley and appraisal tool designed for review purpose are successfully utilized.

The paper includes a detailed description of the tool and summaries of research results. The adopted tabular form is correct, but the publication may be limited to a short description of the tool and a synthetic presentation with commentary on the results without the need to include detailed data (as in Table 2). The analysis of the results are complete.

Detailed comments:

Line 153: Figure 2 should be described as a table. In addition, it is not necessary to include the entire tool, the  shortened comment is enough.

Line 181- “Table 1Error! Reference 181 source not found .. ”- to be corrected

Line 222 (“Table 1Error! Reference source 222 not found.) – to be corrected

The following parts of Table 1 (Lines 183-190), on the following pages, should still be described as Table 1. The Table 2 is later in the publication.

Line 252: (see Table - no number)

VOS Viewer can be taken into consideration to improve the graphical presentation of the results.

Reviewer 3 Report

The manuscript described human biomonitoring (HBM) data in health risk assessments (HRAs). The authors compared the available published data between 2016 and 2021. Thus, these findings will be useful for the acquirement of HBM information. Therefore, the manuscript is not too excellent to be published. In other words, the manuscript is so excellent that it should be published.

Comments

(1) In the first place, what are HBM and HRA?

(2) Why did the number of documents about HBM increase recently in Figure 1?

(3) In line of 321, where is “p. 25”? Moreover, (p. 689) and (p. 699)?

That is all.

Reviewer 4 Report

The authors had carefully reviewed recent advances in the knowledge of human biomonitoring (HBM) data in health risk assessments (HRA) published in peer-reviewed journals between 2016 and 2021. An extensive review of PubMed and Scopus data was carried out. 36 published articles were wisely selected, successfully retrieved, and included in their objective appraisal. Based on their results, the application of HRA theory and practice in the HBM area is not consistent. They recommended some points which need to be considered, to improve the risk-informing potential of HRAs that use HBM data. However, there are some specific points that you would like to consider to improve your manuscript:

  1. Line 51; ‘decisions made’ replace to decisions-maker.
  2. Line 53:65; Please support this sentence with literature reporting epidemiologic, toxicological, and pharmacokinetic data.
  3. Figure 1; ‘AND’ replace to and in the figure and its caption.
  4. Line 74; I think the fundamental elements of HRA should be acknowledged in the introduction section.
  5. Figures 2 and 3; These figures should be improved.
  6. Line 252; see Table??.
  7. Figure 4; it will be better if you replace the white color in this figure with another color.

Author Response

This manuscript is a resubmission of an earlier submission. The following is a list of the peer review reports and author responses from that submission.